# LEARNING TO NAVIGATE THE WEB

**Izzeddin Gur, Ulrich Rueckert, Aleksandra Faust, Dilek Hakkani-Tur**
Google AI
`izzeddingur@cs.ucsb.edu, {rueckert,faust}@google.com, dilek@ieee.org`

## ABSTRACT

Learning in environments with large state and action spaces, and sparse rewards, can hinder a Reinforcement Learning (RL) agent's learning through trial-and-error. For instance, following natural language instructions on the Web (such as booking a flight ticket) leads to RL settings where input vocabulary and number of actionable elements on a page can grow very large. Even though recent approaches improve the success rate on relatively simple environments with the help of human demonstrations to guide the exploration, they still fail in environments where the set of possible instructions can reach millions. We approach the aforementioned problems from a different perspective and propose guided RL approaches that can generate unbounded amount of experience for an agent to learn from. Instead of learning from a complicated instruction with a large vocabulary, we decompose it into multiple sub-instructions and schedule a curriculum in which an agent is tasked with a gradually increasing subset of these relatively easier sub-instructions. In addition, when the expert demonstrations are not available, we propose a novel meta-learning framework that generates new instruction following tasks and trains the agent more effectively. We train DQN, deep reinforcement learning agent, with Q-value function approximated with a novel QWeb neural network architecture on these smaller, synthetic instructions. We evaluate the ability of our agent to generalize to new instructions on World of Bits benchmark, on forms with up to 100 elements, supporting 14 million possible instructions. The QWeb agent outperforms the baseline without using any human demonstration achieving $100\%$ success rate on several difficult environments.

## 1 INTRODUCTION

We study the problem of training reinforcement learning agents to navigate the Web (*navigator agent*) by following certain instructions, such as *book a flight ticket* or *interact with a social media web site*, that require learning through large state and action spaces with sparse and delayed rewards. In a typical web environment, an agent might need to carefully navigate through a large number of web elements to follow highly dynamic instructions formulated from large vocabularies. For example, in the case of an instruction "Book a flight from WTK to LON on 21-Oct-2016", the agent needs to fill out the origin and destination drop downs with the correct airport codes, select a date, hit submit button, and select the cheapest flight among all the options. Note the difficulty of the task: The agent can fill-out the first three fields in any order. The options for selection are numerous, among all possible airport / date combination only one is correct. The form can only be submitted once all the three fields are filled in. At that point the web environment / web page changes, and flight selection becomes possible. Then the agent can select and book a flight. Reaching the true objective in these tasks through trial-and-error is cumbersome, and reinforcement learning with the sparse reward results in the majority of the episodes generating no signal at all. The problem is exacerbated when learning from large set of instructions where visiting each option could be infeasible. As an example, in the flight-booking environment the number of possible instructions / tasks can grow to more than 14 millions, with more than 1700 vocabulary words and approximately 100 web elements at each episode.

A common remedy for these problems is guiding the exploration towards more valuable states by learning from human demonstrations and using pretrained word embeddings. Previous work (Liu et al. (2018); Shi et al. (2017)) has shown that the success rate of an agent on Web navigation

**Instruction: { from: WTK,   to: LON,   date: 10/21/2016 }**

Figure 1: Curriculum learning. Final state is used as the initial state for training the navigator agent. The navigator is assigned only with the easier sub-instruction {to: "LON"}. We detail this process in Section 4.3.

tasks (Miniwob (Shi et al. (2017))) can be improved via human demonstrations and pretrained word embeddings; however, they indeed use separate demonstrations for each environment and as the complexity of an environment increases, these methods fail to generate any successful episode (such as flight booking and social media interaction environments). But in environments with large state and action spaces, gathering the human demonstrations does not scale, as the training needs large number of human demonstrations for each environment.

In this work, we present two methods for reinforcement learning in large state and action spaces with sparse rewards for the web navigation. First, when expert demonstrations or an instruction-following policy (ORACLE) are available, we develop *curriculum-DQN*, a curriculum learning that guides the exploration by starting with an easier instruction following task and gradually increasing the difficulty over a number of training steps. Curriculum-DQN decomposes an instruction into multiple sub-instructions and assigns the web navigation agent (navigator) with an easier task of solving only a subset of these sub-instructions (Figure 1). An expert instruction-following policy (ORACLE) places the agent and goal closer to each other.

Second, when demonstrations and ORACLE policies are not available, we present a novel *meta-learning* framework that trains a generative model for expert instruction-following demonstrations using an arbitrary web navigation policy *without instructions*. The key insight here is that we can treat an arbitrary navigation policy (e. g. random policy) as if it was an expert instruction-following policy for some hidden instruction. If we recover the underlying instruction, we can autonomously generate new expert demonstrations, and use them to improve the training of the navigator. Intuitively, generating an instruction from a policy is easier than following an instruction, as the navigator does not need to interact with a dynamic web page and take complicated actions. Motivated by these observations, we develop an instructor agent, a *meta-trainer*, that trains the navigator by generating new expert demonstrations.

In addition to the two trainers, *curriculum-DQN* and *instructor meta-trainer*, the paper introduces two novel neural network architectures for encoding web navigation Q-value functions, QWeb and INET, combining self-attention, LSTMs, and shallow encoding. QWeb serves as Q-value function for the learned instruction-following policy, trained with either *curriculum-DQN* or *instructor* agent. The INET is Q-value function for the instructor agent. We test the performance of our approaches on a set of Miniwob and Miniwob++ tasks (Liu et al. (2018)). We show that both approaches improve upon a strong baseline and outperform previous state-of-the-art.

While we focus on the Web navigation, the methods presented here, automated curriculum generation with attention-equipped DQN, might be of interest to the larger task planning community working to solve goal-oriented tasks in large discrete state and action Markov Decision Processes.

## 2 RELATED WORK

Our work is closely related to previous works on training reinforcement learning policies for navigation tasks. Shi et al. (2017) and Liu et al. (2018) both work on web navigation tasks and aim to leverage human demonstrations to guide the exploration towards more valuable states where the fo-

cus of our work is investigating the potential of curriculum learning approaches and building a novel framework for meta-training of our deep Q networks. Compared to our biLSTM encoder for encoding the Document Object Model (DOM) trees, a hierarchy of web elements in a web page, Liu et al. (2018) encodes them by extracting spatial and hierarchical features. Diagonal to using pretrained word embeddings as in (Liu et al. (2018)), we used shallow encodings to enhance learning semantic relationships between DOM elements and instructions. Shah et al. (2018) also utilizes an attention-based DQN for navigating in home environments with visual inputs. They jointly encode visual representation of the environment and natural language instruction using attention mechanisms and generate Q values for a set of atomic actions for navigating in a 3D environment.

Modifying the reward function (Ng et al. (1999); Abbeel & Ng (2004)) is one practice that encourages the agent to get more dense reward signals using potentials which also motivated our augmented rewards for Web navigation tasks. Curriculum learning methods (Bengio et al. (2009); Zaremba & Sutskever (2014); Graves et al. (2017)) are studied to divide a complex task into multiple small sub-tasks that are easier to solve. Our curriculum learning is closely related to (Florensa et al. (2017)) where a robot is placed closer to a goal state in a 3D environment and the start state is updated based on robot's performance.

Meta-learning is used to exploit past experiences to continuously learn on new tasks (Andrychowicz et al. (2016); Duan et al. (2016); Wang et al. (2016)). Frans et al. (2018) introduced a meta-learning approach where task-specific policies are trained in a multi-task setup and a set of primitives are shared between tasks. Our meta-learning framework differs from previous works where we generate instruction and goal pairs to set the environment and provide dense rewards for a low level navigator agent to effectively train.

## 3 SETUP

We train a RL agent using DQN (Mnih et al. (2015)) that learns a value function $Q(s, a)$ which maps a state $s$ to values over the possible set of actions $a$. At each time step, the agent observes a state $s_t$, takes an action $a_t$, and observes a new state $s_{t+1}$ and a reward $r_t = r(s_{t+1}, a_t)$. DQN aims to maximize the sum of discounted rewards $\sum_t \gamma^t r_t$ by rolling out episodes as suggested by $Q(s, a)$ and accumulating the reward. We particularly focus on the case where the reward is sparse and only available at the end of an episode. More specifically, for only a small fraction of episodes that are successful, the reward is $+1$; in other cases it is $-1$. This setup combined with large state and action spaces make it difficult to train a Q learning model that can successfully navigate in a Web environment.

In this work, we further make the following assumption where we are given an instruction, $I = [F = (K, V)]$, as a list of fields $F$ where each field is represented as a key-value pair $(K, V)$ (ex. {*from*: "San Francisco", *to*: "LA", *date*: "12/04/2018"}). At each time step, the state of the environment $s_t$ consists of the instruction $I$ and a representation of the web page as a tree $D_t$ of DOM elements (DOM tree). Each DOM element is represented as a list of named attributes such as *tag*, *value*, *name*, *text*, *id*, *class*. We also assume that the reward of the environment is computed by comparing the final state of an episode ($D_N$) with the final goal state $G(I)$. Following Liu et al. (2018), we constrain the action space to `Click(e)` and `Type(e, y)` actions where $e$ is a leaf DOM element in the DOM tree and $y$ is a value of a field from the instruction. Both of these composite actions are mostly identified by the DOM element ($e$), e.g., a *text box* is typed with a sequence whereas a *date picker* is clicked. This motivates us to represent composite actions using a hierarchy of atomic actions defined by the dependency graph in Figure 2.

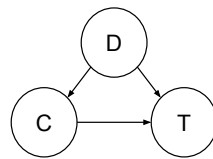

Figure 2: Action dependency graph for hierarchical Q learning. D denotes picking a DOM element, C denotes a click or type action, and T denotes generating a type sequence.

Following this layout, we define our composite Q value function by modeling each node in this graph considering its dependencies :

$$Q(s, a) = Q(s, a_D) + Q(s, a_C | a_D) + Q(s, a_T | a_D, [a_C == "type"]) \qquad (1)$$

where $a = (a_D, a_C, a_T)$ is the composite action, $a_D$ denotes selecting a DOM element, $a_C | a_D$ denotes a "click" or "type" action on the given DOM element, and $a_T | a_D, [a_C == "type"]$ denotes "typing a sequence from instruction" on the given DOM element. When executing the policy (during

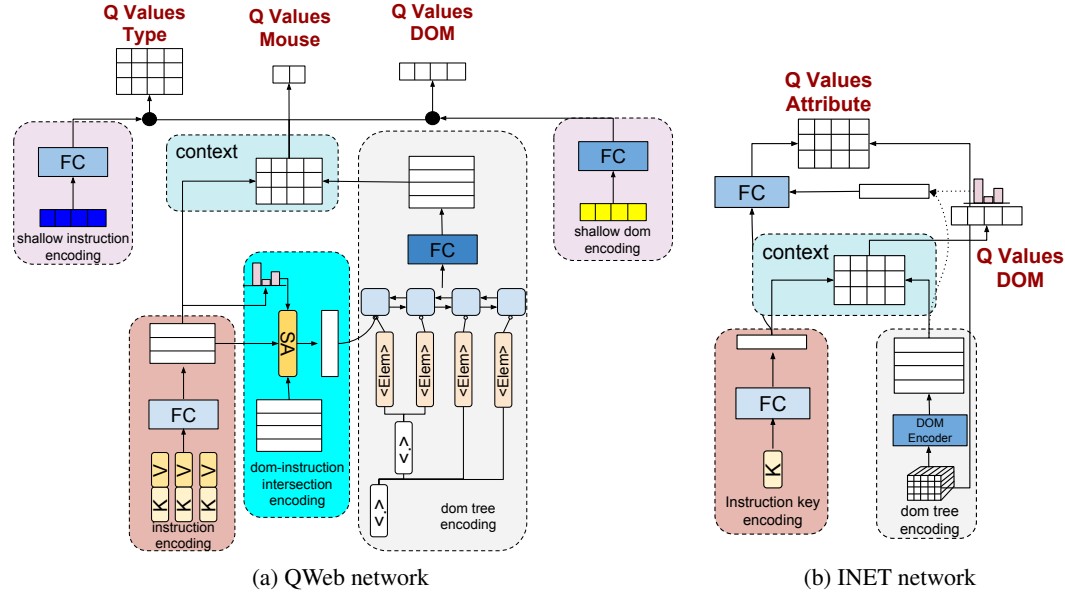

Figure 3: Network architectures a) QWeb and b) INET with attribute pointing. Boxes indicate fully connected layers (FC) with ReLU (for instruction encoding) or tanh (for shallow encoding) activation. (K, V) indicates embeddings of key and value pairs on the instruction; *<Elem>* shows the leaf DOM element embeddings. SA denotes a self-attention mechanism that generates a distribution over the instruction fields. Black circles indicate the gating mechanism to join Q values generated by shallow and deep encodings.

exploration or during testing), the agent first picks a DOM element with the highest $Q(s, a_D)$; then decides to `Type` or `Click` on the chosen DOM element based on $Q(s, a_C | a_D)$; and for a type action, the agent selects a value from the instruction using $Q(s, a_T | a_D, [a_C == "type"])$.

## 4 GUIDED Q LEARNING FOR WEB NAVIGATION

We now describe our proposed models for handling large state and action spaces with sparse rewards. We first describe our deep Q network, called QWeb, for generating Q values for a given observation $(s_t = (I, D_t))$ and for each atomic action $a_D, a_C, a_T$. Next, we explain how we extend this network with shallow DOM and instruction encoding layers to mitigate the problem of learning a large number of input vocabulary. Finally, we delve into our reward augmentation and curriculum learning approaches to solve the aforementioned problems.

---

**Algorithm 1** Curriculum-DQN.

---

   **Input:** ORACLE policy
   **Input:** Sampling probability $p$ and decay rate $d$
   **Web Environment,** $ENV_W$
   **for** number of training steps **do**
      $(I, D, G) \leftarrow$ Sample instruction, initial, and goal states from environment.
      *//Warm-starting the episode.*
      **for** each DOM element $e$ in $D$ **do**
         $\hat{D} \leftarrow$ Get new state from ORACLE$(I, e)$ with probability $p$.
      **end for**
      *// Collect experience and train QWeb starting from the final state $\hat{D}$.*
      Run Algorithm 2 (QWeb, $I, \hat{D}, ENV_W$)
      Decrease $p$ by a factor of $d$
   **end for**

---

### 4.1 DEEP Q NETWORK FOR WEB NAVIGATION (QWEB)

QWeb is composed of three different layers linked in a hierarchical structure where each layer encodes a different component of a given state (Figure 3a): (i) Encoding user instruction, (ii) Encoding

---

**Algorithm 2** One-step DQN training.

**Input:** DQN
**Input:** Goal state, $G$
**Input:** Set of initial states, $\hat{S}$
**Input: Environment,** $ENV$
*//Run DQN policy to collect experience.*
**for** $S \in \hat{S}$ **do**
  $S_i \leftarrow S$
  **while** environment is not terminated **do**
    $o \leftarrow$DQN$(G, S_i)$ to get logits over actions.
    $a \leftarrow$Sample from a categorical distribution $Cat(o)$.
    $(S_{i+1}, R_i) \leftarrow$ Run action $a$ in the environment and collect new state and reward.
    Add $(S_i, a, s_{i+1}, R_i)$ to replay buffer.
    $S_i \leftarrow S_{i+1}$
  **end while**
**end for**
*// Update DQN parameters.*
**for** number of updates per episode **do**
  Sample experience batch from replay buffer.
  Update DQN parameters by running a single step of gradient descent.
**end for**

---

**Algorithm 3** Meta-learning for training QWeb.

instruction generation environment, $ENV_I$
*// Training instructor agent INET.*
**for** number of training steps **do**
  $(G, K_0, I) \leftarrow$Sample goal (a DOM tree), initial state, and instruction from $ENV_I$.
  Run Algorithm 2 $(INET, I, (K_0, G), ENV_I)$
**end for**
Fix the parameters of $INET$.
*// QWeb meta-training via task generation.*
**for** number of training steps **do**
  *// Generate new instruction following task.*
  $(G, P) \leftarrow$ Run $RRND()$ and get goal state and sequence of state, $G$, action pairs, $P$.
  Using $P$ get a dense reward $\hat{R}$.
  $I \leftarrow$ Run $INET(G)$ and get instruction $I$.
  Set up the $ENV_{DR}$ with new task $(I, G)$ and dense rewards.
  $D_0 \leftarrow$ Sample an initial state from the web environment.
  *// Collect experience and train QWeb in environment with dense rewards $ENV_{DR}$.*
  Run Algorithm 2 (QWeb, $I, D_0, ENV_{DR}$)
**end for**

---

the overlapping words between attributes of the DOM elements and instruction, and (iii) Encoding DOM tree. Given an instruction $I = [F = (K, V)]$, QWeb first encodes each field $F$ into a fixed length vector by learning an embedding for each $K$ and $V$. The sequence of overlapping words between DOM element attributes and instruction fields is encoded into a single vector to condition each element on contextually similar fields. Finally, the DOM tree is encoded by linearizing the tree structure (Vinyals et al. (2015)) and running a bidirectional LSTM (biLSTM) network on top of the DOM elements sequence. Output of the LSTM network and encoding of the instruction fields are used to generate Q values for each atomic action.

**Encoding User Instructions** We represent an instruction with a list of vectors where each vector corresponds to a different instruction field. A field is encoded by encoding its corresponding key and value and transforming the combined encodings via a fully connected layer ($FC$) with $ReLU$ activation. Let $E_K^f(i, j)$ ($E_V^f(i, j)$) denote the embedding of the $j$-th word in the key (value) of $i$-th field. We represent a key or value as the average of these embeddings over the corresponding words, i.e., $E_K^f(i) = \frac{1}{|F(i)|} \sum_j E_K^f(i, j)$ represents the encoding of a key. Encoding of a field is then computed as follows : $E^f(i) = FC([E_K(i), E_V(i)])$ where $[.]$ denotes vector concatenation.

**Encoding DOM-Instruction Intersection** For each field in the instruction and each attribute of a DOM element, we generate the sequence of overlapping words. By encoding these sequences in parallel, we generate instruction-aware DOM element encodings. We average the word embeddings over each sequence and each attribute to compute the embedding of a DOM element conditioned on each instruction field. Using a self-attention mechanism, we compute a probability distribution over instruction fields and reduce this instruction-aware embedding into a single DOM element encoding. Let $E(f, D_t(i))$ denote the embedding of a DOM element conditioned on a field $f$ where $D_t(i)$ is the $i$-th DOM element. Conditional embedding of $D_t(i)$ is the weighted average of these embeddings, i.e., $E_C = \sum_f p_f * E(f, D_t(i))$ where self-attention probabilities are computed as $p_f = softmax_i(u * E^f)$ with $u$ being a trainable vector.

**Encoding DOM Trees** We represent each attribute by averaging its word embeddings. Each DOM element is encoded as the average of its attribute embeddings. Given conditional DOM element encodings, we concatenate these with DOM element embeddings to generate a single vector for each DOM element. We run a bidirectional LSTM (biLSTM) network on top of the list of DOM element embeddings to encode the DOM tree. Each output vector of the biLSTM is then transformed through a FC layer with tanh to generate DOM element representations.

**Generating Q Values** Given encodings for each field in the instruction and each DOM element in the DOM tree, we compute the pairwise similarities between each field and each DOM element to generate a context matrix $M$. Rows and columns of $M$ show the posterior values for each field and each DOM element in the current state, respectively. By transforming through a FC layer and summing over the rows of $M$, we generate Q values for each DOM element, i.e., $Q(s_t, a_t^D)$. We use the rows of $M$ as the Q values for typing a field from the instruction to a DOM element, i.e., $Q(s_t, a_t^T) = M$. Finally, Q values for *click* or *type* actions on a DOM element are generated by transforming the rows of $M$ into 2 dimensional vectors using another FC layer, i.e., $Q(s_t, a_t^C)$. Final Q value for a composite action $a_t$ is then computed by summing these Q values : $Q(s_t, a_t) = Q(s_t, a_t^D) + Q(s_t, a_t^T) + Q(s_t, a_t^C)$.

**Incorporating Shallow Encodings** In a scenario where the reward is sparse and input vocabulary is large, such as in flight-booking environments with hundreds of airports, it is difficult to learn a good semantic similarity using only word embeddings. We augment our deep Q network with shallow instruction and DOM tree encodings to alleviate this problem. A joint shallow encoding matrix of fields and elements is generated by computing word-based similarities (such as jaccard similarity, binary indicators such as subset or superset) between each instruction field and each DOM element attribute. We also append shallow encodings of siblings of each DOM element to explicitly incorporate DOM hieararchy. We sum over columns and rows of the shallow encoding matrix to generate a shallow input vector for DOM elements and instruction fields, respectively. These vectors are transformed using a FC layer with tanh and scaled via a trainable variable to generate a single value for a DOM element and a single value for an instruction field. Using a gating mechanism between deep Q values and shallow Q values, we compute final Q values as follows:

$$\hat{Q}(s, a) = Q_{deep}(s_t, a_t^D)(1 - \sigma(u)) + Q_{shallow}(s, a_t^D)(\sigma(u)) \tag{2}$$

$$\hat{Q}(s, a) = Q_{deep}(s_t, a_t^T)(1 - \sigma(v)) + Q_{shallow}(s, a_t^T)(\sigma(v)) \tag{3}$$

where $u$ and $v$ are scalar variables learned during training.

## 4.2 Reward Augmentation

Following Ng et al. (1999), we use potential based rewards for augmenting the environment reward function. Since the environment reward is computed by evaluating if the final state is exactly equal to the goal state, we define a potential function ($Potential(s, g)$) that counts the number of matching DOM elements between a given state ($s$) and the goal state ($g$); normalized by the number of DOM elements in the goal state. Potential based reward is then computed as the scaled difference between two potentials for the next state and current state:

$$R_{potential} = \gamma(Potential(s_{t+1}, g) - Potential(s_t, g)) \tag{4}$$

where $g$ is the goal state. For example, in the flight-booking environment, there are 3 DOM elements that the reward is computed from. Let's assume that at current time step the agent correctly enters the date. In this case, the potential for the current state will increase by $1/3$ compared to the potential of the previous state and the agent will receive a positive reward. Keep in mind that not all actions generate a non-zero potential difference. In our example, if the action of the agent is to click on the date picker but not to choose a specific date, the potential will remain unchanged.

## 4.3 Curriculum Learning

We perform curriculum learning by decomposing an instruction into multiple sub-instructions and assigning the agent with an easier task of solving only a subset of these sub-instructions. We practiced two different curriculum learning strategies to train QWeb: (i) Warm-starting an episode and (ii) Simulating sub-goals.

**Warm-Start.** We warm-start an episode by placing the agent closer to the goal state where the agent only needs to learn to perform a small number of sub-instructions to successfully finish the episode (illustrated in Algorithm 1). We independently visit each DOM element with a certain probability $p$ and probe an ORACLE policy to perform a correct action on the selected element. The environment for the QWeb is initialized with the final state of the Warm-Start process and the original goal of the environment is kept the same. This process is also illustrated in Figure 1 for the flight-booking environment. In this example scenario, QWeb starts from the partially filled web

form (origin and departure dates are already entered) and only tasked with learning to correctly enter the destination airport. At the beginning of the training, we start $p$ with a large number (such as $0.85$) and gradually decay towards $0.0$ over a predefined number of steps. After this limit, the initial state of the environment will revert to the original state of plain DOM tree with full instruction.

**Goal Simulation.** We simulate simpler but related sub-goals for the agent by constraining an episode to a subset of the elements in the environment where only the corresponding sub-instructions are needed to successfully finish an episode. We randomly pick a subset of elements of size $K$ and probe the ORACLE to perform correct set of actions on this subset to generate a sub-goal. The goal of the environment for QWeb will be assigned with the final state of this process and the initial state of the environment will remain unchanged. QWeb will receive a positive reward if it can successfully reach to this sub-goal. At the beginning of the training, we start $K$ with $1$ and gradually increase it towards the maximum number of elements in the DOM tree over a number of steps. After this limit, the environment will revert to the original environment as in warm-start approach.

## 5   META-TRAINER FOR TRAINING QWEB

To address situations when the human demonstrations and ORACLE policy are not available, we combine the curriculum learning and reward augmentation under a more general unified framework, and present a novel meta-training approach that works in two phases. First, we learn to generate new instructions with a DQN agent, *instructor*. *Instructor* learns to recover instructions implied by a non-expert policy (e.g. rule-based or random). We detail the instructor agent, its training environment and neural network architecture in Section 5.1. Second, once the *instructor* is trained, we fix it and use it to provide demonstrations for the QWeb agent from a simple, non-expert, rule-based policy. This process, including the rule-based policy is described in Section 5.2. The whole training process is outlined in Algorithm 3.

### 5.1   LEARNING INSTRUCTIONS FROM GOAL STATES

**Instruction Generation Environment** We define an *instruction state* by the sampled goal and a single *key (K)* sampled without replacement from the set of possible keys predefined for each environment. *Instruction actions* are composite actions: *select a DOM element as in web navigation environments* $(\hat{a}_t^D)$ and *generate a value that corresponds to the current key (K)*, $(\hat{a}_t^K)$. For example, in flight-booking environment, the list of possible keys are defined by the set {*from, to, date*}. Let's assume that the current key is *from* and the agent selected the *departure* DOM element. A value can be generated by copying one of the attributes of the departure element, e.g., *text : "LA"*. After each action, agent receives a positive *reward* (+1) if the generated value of the corresponding key is correct; otherwise a negative reward (-1). To train the *instructor*, we use the same sampling procedure used in the curriculum learning (Algorithm 2 to sample new pairs, but states, actions, and rewards in instruction generation environments are set up differently).

**Deep Q Network for Instruction Generation (INET)** We design a deep Q network, called INET and depicted in Figure 3b, to learn a Q value function approximation for the instruction generation environment. Borrowing the DOM tree encoder from QWeb, INET generates a vector representation for each DOM element in the DOM tree using a biLSTM encoder. Key in the environment state is encoded similarly to instruction encoding in QWeb where only the key is the input to the encoding layer. Q value for selecting a DOM element is then computed by learning a similarity between the key and DOM elements, i.e., $Q^I(s_t, \hat{a}_t^D)$ where $Q^I$ denotes the Q values for the instructor agent. Next, we generate a probability distribution over DOM elements by using the same similarity between the key and DOM elements; and reduce their encodings into a single DOM tree encoding. Q values for each possible attribute is generated by transforming a context vector, concatenation of DOM tree encoding and key encoding, into scores over the possible set of DOM attributes, i.e., $Q^I(s_t, \hat{a}_t^K)$. Final Q values are computed by combining the two Q values : $Q^I(s_t, a_t) = Q^I(s_t, \hat{a}_t^D) + Q^I(s_t, \hat{a}_t^K)$.

### 5.2   META-TRAINING OF THE QWEB (METAQWEB)

We design a simple rule-based randomized policy (RRND) that iteratively visits each DOM element in the current state and takes an action. If the action is `Click(e)`, the agent on click on the

element, and the process continues. If the DOM element is part of a group, that their values depend on the state of other elements in the group (such as radio buttons), RRND clicks on one of them randomly and ignores the others. However, if the action is `Type(e, t)`, the type sequence is randomly selected from a given knowledge source (KS). Consider the flight-booking environment as an example, if the visited element is a text box, RRND randomly picks an airport from a list of available airports and types it into the text box. RRND stops after each DOM element is visited and final DOM tree (D) is generated.

Using pretrained INET model, we generate an instruction $I$ from $D$ and set up the web navigation environment using $(I, D)$ pair. After QWeb takes an action and observes a new state in the web navigation environment, new state is sent to the meta-trainer to collect a meta-reward ($R1$). Final reward is computed by adding $R1$ to the environment reward, i.e., $R = R1 + R2$.

**Discussion.** Even though we use a simple rule-based policy to collect episodes, depending on the nature of an environment, a different kind of policy could be designed to collect desired final states. Also, note that the generated goal states ($D$) need not be a valid goal state. MetaQWeb can still train QWeb by leveraging those incomplete episodes as well as the instruction and goal pairs that the web navigation environment assigns. There are numerous other possibilities to utilize MetaQWeb to provide various signals such as generating supervised episodes and performing behavioral cloning, scheduling a curriculum from the episodes generated by meta-trainer, using MetaQWeb as a behavior policy for off-policy learning, etc. In this work, we practiced using potential-based dense rewards ($R1$) and leave the other cases as future work.

## 6 EXPERIMENTAL RESULTS

We evaluate our approaches on a number of environments from Miniwob (Shi et al. (2017)) and Miniwob++ (Liu et al. (2018)) benchmark tasks. We use a set of tasks that require various combinations of clicking and typing to set a baseline for the QWeb including a difficult environment, *social-media-all*, where previous approaches fail to generate any successful episodes. We then conduct extensive experiments on a more difficult environment, *book-flight-form* (a clone of the original book-flight environment with only the initial web form with a single month is used), that require learning through a large number of states and actions. Each task consists of a structured instruction (also presented in natural language) and a 160px $x$ 210px environment represented as a DOM tree. We cap the number of DOM elements at 100 and the number of fields is 3 for *book-flight-form* environment. Hence, the number of possible actions and number of variables in the state can reach up to 300 and 600, respectively. However, these numbers only reflect a sketch and do not reflect a realization of DOM elements or instruction fields. With more than 700 airports, these numbers greatly increase and generate even larger spaces. *Social-media-all* environment has more than 7000 different possible values in instructions and DOM elements and the task length is 12 which are both considerably higher than other environments. All the environments return a sparse reward at the end of an episode with (+1) for successful and (-1) for failure episodes, respectively. We also use a small step penalty (-0.1) to encourage QWeb to find successful episodes using as small number of actions as possible.

As evaluation metric, we follow previous work and report *success rate* which is the percentage of successful episodes that end with a +1 reward. For instruction generation tasks, *success rate* is +1 if all the values in the instruction is correctly generated.

**Previous Approaches:** We compare the performance of QWeb to previous state-of-the-art approaches. First, SHI17 (Shi et al. (2017)) pre-trains with behavioral cloning using approximately 200 demonstrations for each environment and fine-tunes using RL.They mainly use raw pixels with several features to represent states. Second, LIU18 (Liu et al. (2018)) uses an alternating training approach where a program-policy and a neural-policy are trained iteratively. Program-policy is trained to generate a high level workflow from human demonstrations. Neural-policy is trained using an actor-critic network by exploring states suggested by program-policy.

### 6.1 PERFORMANCE ON MINIWOB ENVIRONMENTS

We evaluate the performance of QWeb on a set of simple and difficult Miniwob environments based on the size of state and action spaces and the input vocabulary. On the simple Miniwob environments (first 11 environments in Figure 4), we show the performance of QWeb without any shallow encoding, reward augmentation or curriculum learning. Figure 4 presents the performance of QWeb

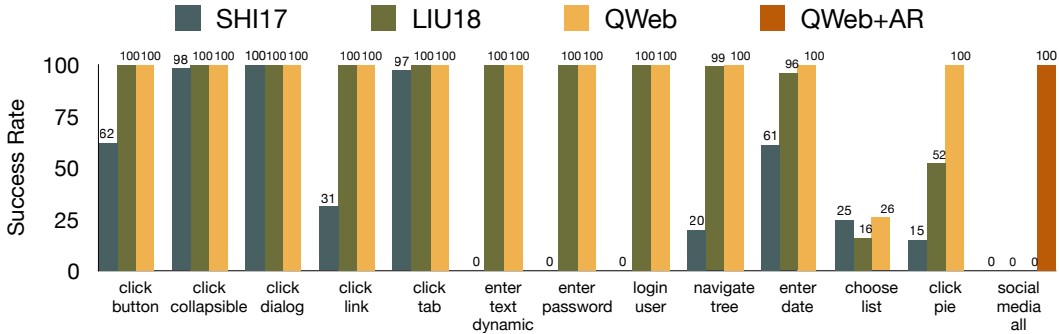

Figure 4: Performance of QWeb on a subset of Miniwob and Miniwob++ environments compared to previous state-of-the-art approaches. AR denotes augmented reward.

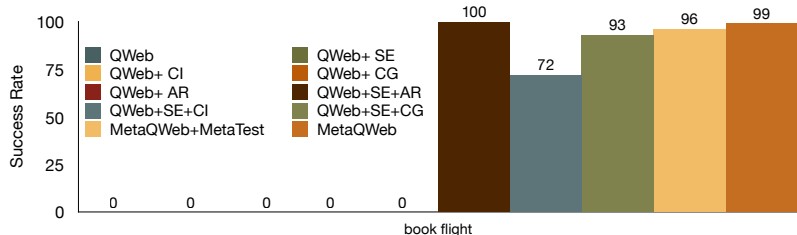

Figure 5: Performance of variants of QWeb and MetaQWeb on the book-flight-form environment. SE, CI, CG, and AR denote shallow encoding, curriculum with warm-start, curriculum with simulated sub-goals, and augmented reward, respectively.

compared to previous approaches. We observe that, QWeb can match the performance of previous state-of-the-art on every simple environment; setting a strong baseline for evaluating our improvements on more difficult environments. We can also loosely confirm the effectiveness of using a biLSTM encoder instead of extracting features for encoding DOM hierarchy where biLSTM encoding gives consistently competitive results on each environment.

Using shallow encoding and augmented rewards, we also evaluate on more difficult environments (*click-pie* and *social-media-all*). On *click-pie* environment, the episode lengths are small (1) but the agent needs to learn a relatively large vocabulary (all the letters and digits in English language) to correctly deduce the semantic similarities between instruction and DOM elements. The main reasons that QWeb is not able to perform well on *social-media-all* environment is that the size of the vocabulary is more than 7000 and task length is 12 which are both considerably larger compared to other environments. Another reason is that the QWeb can not learn the correct action by focusing on a single node; it needs to incorporate siblings of a node in the DOM tree to generate the correct action. In both of these environments, QWeb+SE successfully solves the task and outperforms the previous approaches without using any human demonstration. Our empirical results suggest that it is critical to use both shallow encoding and augmented rewards in *social-media-all* environment where without these improvements, we weren't able to train a successful agent purely from trial-and-error. Without augmented reward, the QWeb overfits very quickly and gets trapped in a bad minima; in majority of these cases the policy converges to terminating the episode as early as possible to get the least step penalty.

We analyze the number of training steps needed to reach the top performance on several easy (*login-user*, *click-dialog*, *enter-password*) and difficult (*click-pie*) environments to understand the sample efficiency of learning from demonstrations (as in LIU18 (Liu et al. (2018))). We observe that using demonstrations can lead to faster learning where on average it takes less than 1000 steps for LIU18 and more than 3000 steps for QWeb (96$k$ steps for *login-user* environment) however with a drop in performance on the more difficult *click-pie* environment: LIU18 reaches 52% success rate at 13$k$ steps while QWeb reaches the same performance at 31$k$ steps and 100% success at 175$k$ steps.

## 6.2 PERFORMANCE ON BOOK FLIGHT ENVIRONMENT

In Figure 5, we present the effectiveness of each of our improvements on *book-flight-form* environment. Without using any of the improvements or using only a single improvement, QWeb is not able to generate any successful episodes. The main reason is that without shallow encoding, QWeb is not able to learn a good semantic matching even with a dense reward; with only shallow encoding vast majority of episodes still produce -1 reward that prevents QWeb from learning. When we analyze the cause of the second case, we observe that as the training continues QWeb starts clicking submit button at first time step to get the least negative reward. When we remove the step penalty or give no reward for an unsuccessful episode, QWeb converges to one of the modes: clicking submit button at first time step or generating random sequence of actions to reach step limit where no reward is given. Using shallow encoding, both curriculum approaches offer very large improvements reaching above 90% success rate. When the reward can easily be augmented with a potential-based dense reward, we get the most benefit and completely solve the task.

Before we examine the performance of MetaQWeb, we first evaluate the performance of INET on generating successful instructions to understand the effect of the noise that MetaQWeb introduces into training QWeb. We observe that INET gets 96% success rate on fully generating an instruction. When we analyze the errors, we see that the majority of them comes from incorrect DOM element prediction (75%). Furthermore, most of the errors are on date field (75%) where the value is mostly copied from an incorrect DOM element. After we train QWeb using meta-training, we evaluate its performance in two different cases : (i) We meta-test QWeb using the instruction and goal pairs generated by MetaQWeb to analyze the robustness of QWeb to noisy instructions and (ii) We test using the original environment. Figure 5 suggests that in both of the cases, MetaQWeb generates very strong results reaching very close to solving the task. When we examine the error cases for meta-test use case, we observe that 75% of the errors come from incorrect instruction generations where more than 75% of those errors are from incorrect date field. When evaluated using the original setup, the performance reaches up to 99% success rate showing the effectiveness of our meta-training framework for training a successful QWeb agent.

The main reason of the performance difference between the QWeb+SE+AR and the MetaQWeb can be explained by the difference between the generated experience that these models learn from. In training QWeb+SE+AR, we use the original and clean instructions that the environment sets at the beginning of an episode. MetaQWeb, however, is trained with the instructions that instructor agent generates. These instructions are sometimes incorrect (as indicated by the error rate of INET : 4%) and might not reflect the goal accurately. These noisy experiences hurt the performance of MetaQWeb slightly and causes the 1% drop in performance.

## 7 CONCLUSION AND FUTURE WORK

In this work, we presented two approaches for training DQN agents in difficult web navigation environments with sparse rewards and large state and action spaces, one in presence of expert demonstrations and the other without the demonstrations. In both cases, we use dense, potential-based rewards to augment the training. When an expert demonstrations are available, curriculum learning decomposes a difficult instruction into multiple sub-instructions and tasks the agent with incrementally larger subset of these sub-instructions; ultimately uncovering the original instruction. When expert demonstrations are not available, we introduced a meta-trainer that generates goal state and instruction pairs with dense reward signals for the QWeb to train more efficiently. Our models outperform previous state-of-the-art models on challenging environments without using any human demonstration. The evaluations also indicate that having a high-quality expert demonstrations is important, as the policies trained from curriculum over demonstrations outperform policies that generate non-perfect demonstrations. In future work, we plan to apply our models on a broader set of navigation tasks with large discrete state and actions, and will experiment with other signals to utilize in the meta-trainer, such as supervised pre-training using behavioral cloning, scheduling a curriculum from the episodes generated by meta-trainer, using meta-trainer as off-policy learning, etc.

### ACKNOWLEDGMENTS

We thank Amir Fayazi for his help with integrating the Miniwob benchmarks into our ecosystem. We are grateful to Pranav Khaitan and the Deep Dialogue team at Google Research for discussions, as well as to the anonymous reviewers for their valuable feedback.

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
