# OpenReview forum: "Learning to Navigate the Web"
_ICLR.cc/2019/Conference_

### Official Review · AnonReviewer1 · 2018-10-19
**A novel proposal addressing a complex problem with a large number of components but without a clear analysis of their relevance**

**Rating:** 7
**Confidence:** 3

**Review:**

The paper propose a framework to deal with large state and action
spaces with sparse rewards in reinforcement learning. In particular,
they propose to use a meta-learner to generate experience to the agent
and to decompose the learning task into simpler sub-tasks. The authors
train a DQN with a novel architecture to navigate the Web.
In addition the authors propose to use several strategies: shallow
encoding (SE), reward shaping (AR) and curriculum learning (CI/CG).
It is shown how the proposed method outperforms state-of-the-art
systems on several tasks.

In the first set of experiments it is clear the improved performance
of QWeb over Shi17 and Liu18, however, it is not clear why QWeb is not
able to learn in the social-media-all problem. The authors tested only
one of the possible variants (AR) of the proposed approach with good
performance.

It is not clear in the book-flight-form environment, why the
QWeb+SE+AR obtained 100% success while the MetaQWeb, which includes
one of main components in this paper, has a lower performance.

The proposed method uses a large number of components/methods, but it
is not clear the relevance of each of them. The papers reads like, "I
have a very complex problem to solve so I try all the methods that I
think will be useful". The paper will benefit from an individual
assessment of the different components.

The authors should include a section of conclusions and future work.

---

> ### Author Response · Authors · 2018-11-26
> **Thank you for the comments and questions.**
>
> We thank the reviewer for the comments and questions. Below are our responses.
>
> > “In the first set of experiments it is clear the improved performance of QWeb over Shi17 and Liu18, however, it is not clear why QWeb is not able to learn in the social-media-all problem. The authors tested only one of the possible variants (AR) of the proposed approach with good performance.”
>
> The main reason is that in social-media-all environment, the size of the vocabulary is more than 7000 and task length is 12 which are both considerably larger compared to other environments. Another reason is that the QWeb can not learn the correct action by focusing on a single node; it needs to incorporate siblings of a node in the DOM tree to generate the correct action. Without adding shallow encoding (SE) and one of the proposed approaches (such as AR), QWeb is not able to train purely from trial-and-error as the number of successful episodes is very small.
>
> We updated the Section 6.1 of the paper with these explanations and we plan to conduct more experiments in future work.
>
> > “It is not clear in the book-flight-form environment, why the QWeb+SE+AR obtained 100% success while the MetaQWeb, which includes one of main components in this paper, has a lower performance.”
>
> The main reason of the performance difference between the QWeb+SE+AR and the MetaQWeb can be explained by the difference between the generated experience that these models learn from. In training QWeb+SE+AR, we use the original and clean instructions that the environment sets at the beginning of an episode. MetaQWeb, however, is trained with the instructions that instructor agent generates. These instructions are sometimes incorrect (as indicated by the error rate of INET : $4\%$) and might not reflect the goal accurately. These noisy experiences hurt the performance of MetaQWeb slightly and causes the $1\%$ drop in performance.
>
> We updated the Section 6.2 with this explanation.
>
> > “The proposed method uses a large number of components/methods, but it is not clear the relevance of each of them. The papers reads like, "I have a very complex problem to solve so I try all the methods that I think will be useful". The paper will benefit from an individual assessment of the different components.”
>
> Thank you for the comment. We have revised the Introduction, and Sections 4 and 5 to clarify the differences between the methods and contributions. Below is the summary that hopefully brings more clarity to the reasoning before the approaches.
>
> We aim to solve the web navigation tasks in two situations, when the expert demonstrations are available and when they are not. When the expert demonstrations are available, we need to make several improvements to the training to outperform the baselines. These improvements are: better neural network architecture (QWeb), and more dense rewards. We get the more dense rewards by using the reward potentials and setting up a curriculum over the given demonstrations.
>
> In the second case, when the expert demonstrations are not available. In that situation, we use the meta-trainer to generate new demonstrations.
>
> > ”The authors should include a section of conclusions and future work.”
> Thank you for point it out. The section is added to the paper.

---

### Official Review · AnonReviewer3 · 2018-10-26
**solid experiments, needs clarity improvement**

**Rating:** 8
**Confidence:** 3

**Review:**

UPDATE:

Thank you to the authors for a comprehensive response.  I have increased my score based on these changes.  I apologize for the misunderstanding about ArXiV papers and indeed the authors are correct on that point.  Thank you as well for reporting the learning speeds.  As you mentioned, they confirm our intuitions and complete the picture of the algorithm’s behavior.  The addition of pseudo-code does make the paper and algorithm easier to follow.  Thank you for adding it.  The rewritten section 5 is indeed much easier to follow and makes the coordination between the agents clear.  Seeing that the instructor is a fixed policy resolves the game theoretic issue form the original review.


Summary:

The paper proposes a deep reinforcement learning approach to filling out web forms, called QWeb.  In addition to both deep and shallow embeddings of the states, the authors evaluate various methods for improving the learning system, including reward shaping, introducing subgoals, and even a meta-learning algorithm that is used as an instructor.  These variations are tested in several environments and basic QWeb is shown to outperform the baselines and many of the adaptations perform even better than that in more complex domains.

Review:

Overall, the problem the paper considers is important and their results seem significant.  The authors have derived a novel architecture and are the first to tackle the problem of filling in web forms at this scale with an autonomous learning agent rather than one that is taught mostly by demonstration.

The related work section is very well written with topical references to recent results and solid differentiations to the new algorithm.  However, I see many references in the paper are not from peer reviewed conferences or journals.  Unless absolutely necessary, such papers should not be cited because they have not been properly peer reviewed.  If the papers cited have actually been in a conference or journal, please add the correct attribution.

The experiments seem well conducted.  I liked that each new addition to the algorithm was tested incrementally in Figure 7 to give a realistic view of the gains introduced by each change.  I also thought the earlier comparisons to the baselines were well done and I liked that they were done against modern cutting-edge LfD demonstrations.  The only thing I would have liked to seen beyond these results are actual learning curves showing, after X iterations, what percentage of the tasks could be completed.  I suspect that in many domains the baseline LfD techniques are learning much faster since learning from teachers tends to be more targeted and sample efficient.  Learning curves would show us whether or not this is the case.

The weakest part of the paper was the description of the instructor network and the Meta-training in general.  This portion seemed ill-described and largely speculative, despite the promising results in Figure 7.  In particular, Section 5 is very unclear on how exactly the Meta-Learning works.  Pseudocode is definitely needed in this portion well beyond the quick descriptions in Figure 4 and 5, which I could not understand, despite multiple readings.  I suggest eliminating those figures and providing concrete pseudo—code describing the meta learning and also addressing the following open questions in the text:
•	Why is a rule based randomized policy good to learn from?  How is this different from learning from demonstration in the baselines?
•	How is a “fine grained signal” generated?  What does that mean?  Is it a reward?
•	In Section 5.1, are there two RL agents, an instructor and a learner with different reward functions?  If so, isn’t this becoming game theoretic and is this likely to converge in most scenarios?
•	What does Q_D^I actually represent?  Why is maximizing these values a good thing?

There are a few grammatical mistakes in the paper including:

Abstract – simpler environments -> simple environments
Abstract- with gradually increasing -> with a gradually increasing
Page 2 – generate unbounded -> generate an unbounded
Page 7 – correct value -> correct values
Page 9 – episode length -> episode lengths

---

> ### Author Response · Authors · 2018-11-26
> **Thank you for the insightful comments (1/2).**
>
> We thank the reviewer for the insightful comments. Below are our responses.
>
> > “However, I see many references in the paper are not from peer reviewed conferences or journals.  Unless absolutely necessary, such papers should not be cited because they have not been properly peer reviewed.”
>
> Thank you for pointing that out. We updated the references where the archival versions became available, and will do so again before camera-ready if accepted. At the same time, we also wanted to kindly point out that ICLR reviewer guidelines consider publication on Arxiv as prior work that should be properly cited: https://iclr.cc/Conferences/2019/Reviewer_Guidelines
>
>
> > “The only thing I would have liked to seen beyond these results are actual learning curves showing, after X iterations, what percentage of the tasks could be completed.  I suspect that in many domains the baseline LfD techniques are learning much faster since learning from teachers tends to be more targeted and sample efficient.  Learning curves would show us whether or not this is the case. “
>
> We collected the number of steps (k=1000) needed to reach the top performance :
> ___________________________________________________
> | environment \ method  |  QWeb   |  LIU18 |
> --------------------------------------------------------------
> |  click-pie                           | 175k      |  13k      |
> |  login-user                       |  96k       |  < 1k     |
> |  click-dialog                     |  5k         |  < 1k     |
> |  enter-password             |  3k         |  < 1k     |
> --------------------------------------------------------------
>
> These numbers reflect the reviewer’s intuition that LfD techniques learn faster, however, with a drop in success rate for some environments. We updated the experimental results in Section 6.1 with these results.
>
> > ”The weakest part of the paper was the description of the instructor network and the Meta-training in general.  This portion seemed ill-described and largely speculative, despite the promising results in Figure 7.  In particular, Section 5 is very unclear on how exactly the Meta-Learning works.  Pseudocode is definitely needed in this portion well beyond the quick descriptions in Figure 4 and 5, which I could not understand, despite multiple readings.  I suggest eliminating those figures and providing concrete pseudo—code describing the meta learning and also addressing the following open questions in the text:”
>
> Thank you so much for the suggestion. We added the Algorithms 1, 2, 3 for the curriculum learning, DQN training, and meta learning. We removed Figure 5, and put Figures 3 and 4 side-by-side, since they both depict neural network architecture. We have also rewrote the Section 5. We are hoping that the changes are improving the clarity.
>
> > “Why is a rule based randomized policy good to learn from?  How is this different from learning from demonstration in the baselines?”
> When the expert demonstrations are not available, we can use any policy (random or rule-based) and pretend that the policy is following some, to us known, instruction. The instructor agent learns to recover that hidden instructions, in effect creating new demonstrations. Once the instructor is trained to recover the instructions for a given policy, we generate new instruction / goal paths so that we can train QWeb. The choice of policy is arbitrary, and it was a design choice to select a simple, rule-based policy that visits each DOM element in web navigation environments.
>
> Our meta-training approach has two main advantages over learning from demonstrations:
> By learning to generate new instruction following tasks, we can generate unbounded amount of episodes for any environment where collecting large amount of episodes for each environment is costly.
> Similar to our curriculum generation with simulated goals approach, generated goal states are allowed to be incomplete. For example, if we constrain our rule based policy to run only a small number of steps, generated goal state could be incomplete and some DOM elements in the web page could be unvisited. In this case, QWeb can still leverage these experiences while also learning from the original instructions and sparse rewards that the environment generates.
>
> Paper’s introduction, and Section 5 are updated to clarify the role and selection of the rule-based policy, and advantages over the baselines.
>
> > “How is a “fine grained signal” generated?  What does that mean?  Is it a reward?”
> Thank you for pointing it out. Yes, it is a dense reward. We updated the paper to use more commonly used term: dense reward.

---

> > ### Author Response · Authors · 2018-11-26
> > **Thank you for the insightful comments (2/2).**
> >
> > > “In Section 5.1, are there two RL agents, an instructor and a learner with different reward functions?  If so, isn’t this becoming game theoretic and is this likely to converge in most scenarios?”
> >
> > There are two different RL agents : instructor agent (INET) and navigator agent (QWeb). These are trained in two phases : (i) we first train INET (a DQN agent with Q value function defined at the end of Section 5.1) using the instruction generation environment that we described in Section 5.1, (ii) next, parameters of the INET agent is fixed and we train QWeb using the instruction and goal pairs that the meta-trainer generates by running INET at the beginning of each episode. Hence, we avoid the problems that could have arised by jointly training two different RL agents with different objectives.
> >
> > > “What does Q_D^I actually represent?  Why is maximizing these values a good thing?”
> > Q_D^I is the Q value function that we used to train instructor agent (INET) as we described in Section 5.1.
> >
> > > “There are a few grammatical mistakes in the paper including.”
> > Thank you for pointing it out. We updated in the paper, and will make another pass for the final version if accepted.

---

### Official Review · AnonReviewer2 · 2018-11-03
**a good RL application paper for dealing with large action and state spaces**

**Rating:** 7
**Confidence:** 3

**Review:**

This paper developed a curriculum learning method for training an RL agent to navigate a web. It is based on the idea of decomposing an instruction in to multiple sub-instructions, which is equivalent to decompose the original task into multiple easy to solve sub-tasks. The paper is well motivated and easily accessible. The problem tackled in this work is an interesting application of RL dealing with large action and state spaces. It also demonstrates superior performance over the state of the art methods on the same domains

Here are the comments for improving this manuscript:

There are a few notations used without definition, for example DOM tree, Potential (in equation (4))

Some justification regarding the the Q value function specified in (1) might be helpful, otherwise it looks very adhoc.

Although using both shallow encoding and augmented reward lead to good empirical results, it might be useful to give more insights, for example, sample size limit cause overfitting for deep models?

What are the sizes of action state and action spaces?

The conclusion part is missing.

---

> ### Author Response · Authors · 2018-11-26
> **Thank you for the kind words and questions that help us improve the paper.**
>
> We thank the reviewer for the kind words and questions that help us improve the paper. We detail our responses below.
>
> > “There are a few notations used without definition, for example DOM tree, Potential (in equation (4))”
>
> We updated in the paper.
> On Page 3, line 3: “the Document Object Model (DOM) trees, a hierarchy of web elements in a web page.”
>
> Section 4.2:“we define a potential function ($Potential(s, g)$) that counts the number of matching DOM elements between a given state (s) and the goal state (g); normalized by the number of DOM elements in the goal state. Potential based reward is then computed as the scaled difference between two potentials for the next state and current state”
>
> > “Some justification regarding the the Q value function specified in (1) might be helpful, otherwise it looks very adhoc”.
>
> Our Q value function in Eq. (1) is motivated by the design of our composite actions (click(e) and type(e, y)) and the nature of web pages in general. A DOM element (e) in a web page mostly identifies which composite action to select, e.g., a text box such as destination airport is typed with the name of an airport code while a date picker is clicked. This motivates the dependency graph that we sketched in Figure 2. We define our Q value function for each composite action based on this dependency graph via a separate value function to model each node in the graph given its dependencies. We also also added this motivation to Section 3.
>
>
> > “Although using both shallow encoding and augmented reward lead to good empirical results, it might be useful to give more insights, for example, sample size limit cause overfitting for deep models?”
>
> We would like to give more insights into overfitting of deep models without and with augmented rewards. Without augmented rewards, the Q function overfits very early to the minimum Q value possible since the majority of the episodes are unsuccessful and the reward is highly unbalanced towards negative. Escaping this bad minima via purely random exploration is difficult especially in environments that require longer episodes. We observe that in majority of these cases the policy converges to terminating the episode as early as possible to get the least step penalty. With augmented rewards, Q function recovers from these cases very quickly and gradually learns from more successful episodes. We also added these insights into Section 6.1.
>
>
> > “What are the sizes of action state and action spaces?”
>
> Our action and state spaces are mainly defined by the number of DOM elements in web pages and number of fields in the instructions. For example, in flight-booking-form environment, the number of DOM elements is capped at 100, number of fields is 3, and there are two types of actions (click or type). Hence, the number of possible actions would reach 600 and number of possible variables in a state reach 300. These numbers, however, do not reflect the possible “realization” of a DOM element or a field; they just reflect a sketch. For example, “from” field can take a value from 700 possible airports or “destination” input DOM element can be repetitively typed with any value from the instruction. These greatly increase the space of both states and actions. We added this description into the Section 6.1.
>
>
> > “The conclusion part is missing.”
> Thank you for pointing that out. We added the conclusion in the paper.

---

### Meta-Review · Area_Chair1 · 2018-12-14
**Important topic, solid contribution**

**Confidence:** 3
**Recommendation:** Accept (Poster)

**Metareview:**

All reviewers (including those with substantial expertise in RL) were solid in their praise for this paper that is also tackling an interesting application that is much less well studied but deserves attention.